# Nutrition and Disorders of Gut–Brain Interaction

**DOI:** 10.3390/nu16010176

**Published:** 2024-01-04

**Authors:** Emidio Scarpellini, Lukas Michaja Balsiger, Bert Broeders, Karen Van Den Houte, Karen Routhiaux, Karlien Raymenants, Florencia Carbone, Jan Tack

**Affiliations:** 1Translational Research in Gastrointestinal Disoerders (T.A.R.G.I.D.), Gasthuisberg University Hospital, KU Leuven, Herestraat 49, 3000 Lueven, Belgium; emidio.scarpellini@med.kuleuven.be (E.S.); lukasmichaja.balsiger@kuleuven.be (L.M.B.); bert.broeders@med.kuleuven.be (B.B.); karen.vandenhoute@med.kuleuven.be (K.V.D.H.); karen.routhiaux@med.kuleuven.be (K.R.); karlien.raymenants@med.kuleuven.be (K.R.); florencia.carbone@uzleuven.be (F.C.); 2Internal Medicine Unit, “Madonna del Soccorso” General Hospital, Via Luciano Manara 7, 63074 San Benedetto del Tronto, Italy

**Keywords:** nutrients, disturbances of gut–brain axis, functional dyspepsia, malabsorption, irritable bowel syndrome, diarrhea, constipation

## Abstract

Background: Disorders of gut–brain interaction (DGBIs) have a complex pathophysiology that is often characterized by a relationship between food ingestion and triggering of symptoms. Understanding of the underlying mechanisms and the role of nutrients as a therapeutic target are rapidly evolving. Aims and methods: We performed a narrative review of the literature using the following keywords, their acronyms, and their associations: nutrients, disorders of gut–brain interaction; functional dyspepsia; malabsorption; irritable bowel syndrome; diarrhea; constipation. Results: Functional dyspepsia displayed a significant correlation between volume, fat and/or wheat abundance, chemical composition of ingested food and symptoms of early satiety, fullness and weight loss. Carbohydrate malabsorption is related to enzyme deficiency throughout the GI tract. Food composition and richness in soluble vs. non-soluble fibers is related to constipation and diarrhea. The elimination of fermentable oligo-, di-, monosaccharides and polyols (FODMAPs) has a significant and non-unidirectional impact on irritable bowel syndrome (IBS) symptoms. Conclusions: Food volume, nutritive and chemical composition, and its malabsorption are associated with symptom generation in DGBIs. Further multicenter, randomized-controlled clinical trials are needed to clarify the underlying pathophysiology.

## 1. Introduction

Disorders of gut–brain interaction (DGBIs), previously known as functional gastrointestinal disorders (FGIDs) recognize within their pathophysiology a significant relationship of symptom occurrence with food ingestion, its progression through the GI tract and food digestion and absorption [1,2].

Over the last two decades, a solid literature emerged which clearly shows the pathophysiological relationship between ingestion of certain foods or food-components, visceral hypersensitivity, altered central nervous system (CNS) processing of these stimuli and triggering or worsening of symptoms [3]. This has led to the development of the low-fermentable oligo-, di-, monosaccharide, and polyols (FODMAP) diet, successfully applied for symptom control in irritable bowel syndrome (IBS) [4]. In parallel, the efficacy of a gluten free diet has been evaluated in the same patient population [5]. A low wheat diet is also in line with this efficacy profile [6]. One challenge with these diets, especially the low FODMAP diet, are their complexity, affecting applicability and compliance in short and long term [1].

Besides in IBS, there is also a well-established role for diet factors in other bowel DGBIs such as functional constipation and diarrhea [1,7,8].

In the upper GI tract, the role of food has only recently emerged as a topic of intense research for its role in disorders such as. gastro-esophageal reflux, functional dyspepsia (namely, epigastric pain and post-prandial distress syndromes) [1]. Early studies on the complex interplay between diet and altered esophago-gastric motility, visceral hyperalgesia and CNS processing and neuromodulation seem promising [9]. This can have a specific and significant impact on future mechanistic and clinical protocols in several disorders of the upper-GI tract. The role of diet as trigger and treatment is best established for celiac disease, a malabsorptive condition triggered by an immune response to wheat protein [10].

Thus, we will review the main DGBIs of the upper and lower GI tract, the pathophysiological link with diet and its components, as well as established and emerging nutritional therapy approaches.

## 2. Materials and Methods

We conducted a search on PubMed and Medline for original articles, reviews, meta-analyses, and case series using the following keywords, their acronyms, and their associations: nutrients, disturbances of the gut–brain axis; functional dyspepsia; malabsorption; irritable bowel syndrome; diarrhea; and constipation. When appropriate, we included preliminary evidence from abstracts from main national and international nutrition and gastroenterological meetings (e.g., European Society for Enteral and Parenteral Nutrition, United European Gastroenterology Week, Digestive Disease Week). The last Medline search was dated 30 September 2023.

In detail, we included 96 papers, as follows: 36 reviews of literature, 8 metanalyses, 51 original papers, one congress abstract. Among the 51 original papers, we retrieved 41 RCTs.

## 3. Results

### 3.1. The Role of Food Intake as a Symptom Trigger and Therapeutic Target of Functional Esophageal Disorders (FEDs)

Functional esophageal disorders have a physiopathology similar to irritable bowel syndrome and functional dyspepsia, as they depend on abnormalities of gut–brain interaction and central nervous system processing of peripheral esophageal stimuli. Patients with these disturbances can be “hypervigilant” about minor symptoms or “hypersensitive” also to physiological amounts of acid refluxed in the esophagus [11]. Patients may present with dysphagia, or, more commonly, chest pain or heartburn [12].

RCTs used red chili for 5–6 weeks in the treatment of patients with dyspeptic symptoms, showing a significant improvement of these complaints and, also of typically GERD-symptoms (namely heartburn) [13]. Chronic esophageal exposure (5–6 weeks) to capsaicin, the active ingredient in red chili, may desensitize the esophagus and modulate the TRPV1 receptors [14], reducing the mucosal “hypersensitivity”. However, these studies had small sample size and were enrolling mainly healthy subjects and non-erosive reflux disease patients.

Indeed, patients suffering from functional esophageal disorders should be taught to avoid an extremely restrictive diet that may be detrimental for their nutritional status. There is lack of solid evidence on avoidance of fatty meals and reduction of weight, but there is a well-established use of these recommendations clinical practice.

### 3.2. The Role of Food Intake as a Symptom Trigger and Therapeutic Target of Functional Dyspepsia

In functional dyspepsia (FD), a disorder that etymologically refers to “bad digestion”, most patients experience worsening of gastrointestinal symptoms after a meal. In the pure postprandial distress syndrome (PDS) and the PDS-epigastric pain syndrome (EPS) overlap subtypes, both accounting for 82% of all FD cases, the presence of symptoms such as early satiation, postprandial distress or meal-related epigastric pain is necessary to fulfill the ROME IV diagnostic criteria for FD-PDS or PDS/EPS-overlap [15]. In a large clinical trial that investigated symptom development before and after a standardized meal, 78% of patients with functional dyspepsia reported aggravation of their symptoms, with postprandial fullness as dominant symptom [16].

#### 3.2.1. Food as a Trigger

Many FD patients link their symptoms to specific foods, report food avoidance or experience symptom worsening postprandially [2]. FD patients consume fewer meals per day and have a lower caloric intake and daily fat intake compared to healthy individuals [17]. Regarding nutrient composition, a systematic analysis of 16 relevant studies identified foods high in fat and gluten as most provocative for dyspeptic symptoms [18]. This is supported by the fact that FD patients commonly report grain/wheat products, takeout foods, and processed foods as the culprit of their complaints [19]. In a large cross sectional (N = 3362) analysis in a Middle Eastern population the daily intake of fruit was inversely correlated to the risk of having early satiation or postprandial fullness [20]. Furthermore, in a meta-analysis, ultra processed food has been associated with an increased risk of FD [21] (Figure 1).

#### 3.2.2. Food Volume as a Trigger

During ingestion of a meal, the gastric accommodation reflex generates a relaxation of the proximal stomach, which provides a reservoir for the storage of ingested food without a rise in intragastric pressure. The increased meal-volume in the stomach stretches mechano-receptors in the gastric wall (increasing wall tension) which will lead to sensations of increasing satiation, finally leading to the termination of the meal. Studies in healthy subjects have demonstrated the importance of impaired gastric accommodation and hypersensitivity to gastric filling in the development of symptoms [22].

In FD, impaired accommodation and hypersensitivity to gastric distention are common pathophysiological features, each seen in 30% to 40% of FD patients in tertiary care. While impaired gastric accommodation is associated with increased prevalence of symptoms of early satiety and weight loss, hypersensitivity to distention in FD was associated with symptoms of postprandial pain, excessive belching and unexplained weight loss [23,24] (Figure 1).

#### 3.2.3. Chemosensing as a Trigger

In addition to mechanical sensing, visceral sensation can be determined by means of transmission of chemosensitive modalities such as nutrient sensing, acid sensitivity, tastant sensing [24].

Capsaicin, the natural compound of chili pepper, is an agonist at the transient receptor potential vanilloid type 1 (TRPV1) channel which converts thermal and chemical stimuli into painful sensations or discomfort [25]. Moreover, it has been previously observed that the TRPV1 channel could also be involved in FD by studies using gastric application of capsaicin in FD patients showing an increase in hypersensitivity and dyspeptic symptoms [26]. Another molecule found in food, menthol, which is a TPRM8 and TRPA1 agonist [27], has been shown to be effective in treating EPS and PDS symptoms in a placebo-controlled trial [28,29] and a large open label trial [18] (Figure 1).

#### 3.2.4. Caloric Content as a Trigger

It has been confirmed by multiple studies that FD patients have a lower liquid meal tolerance capacity than healthy volunteers associated with impaired gastric accommodation [30,31]. Meal-related symptoms due to lipid or calory rich food cannot only be explained by delayed gastric emptying or decreased gastric accommodation, as infusion of nutrients directly into the duodenum has also been shown to provoke dyspepsia symptoms in FD patients [32].

Not only composition of food, such as caloric or fat content, determines satiety and bloating scores in FD, but also cognitive factors play a role [33]. Information given about the amount of fat content can induce higher satiety or bloating scores, even if the same meal is consumed (Figure 1).

#### 3.2.5. Food Allergens as a Trigger

Currently there is emerging evidence of a central role of duodenal permeability changes associated with low-grade eosinophilic and mast cell inflammation [34] and activation [35]. These findings lead to the hypothesis of a potential allergic reaction to certain food proteins, similarly to what is observed in eosinophilic esophagitis (EoE) [36]. Although physicians advise patients with dyspeptic symptoms to adjust their diet, there is little scientific evidence on the impact of dietary habits in management of FD. Nevertheless, the general clinical advice to patients is to frequently eat small meals and avoid high-fat food [37].

The effect of previously mentioned standard dietary advice has shown to be similar to the effect of a low FODMAP diet in FD [38]. Moreover, symptoms assessed by the LPDS diary and impaired mucosal integrity have been shown to improve after a 6-week strict low FODMAP diet and both aspects were associated [39]. A recent report on the six-food elimination diet in FD showed significant symptom improvement without changes in duodenal eosinophil counts or duodenal permeability. Finally, a case–control study found that gluten-containing foods might trigger symptoms of FD such as early satiety and in a large Australian population-based cohort study, self-reported wheat sensitivity was associated with FD and IBS [40,41]. The culprit food items are still unclear based on the limited available literature. Further future high quality dietary intervention studies are required to establish evidence based dietary treatment in FD (Figure 1).

### 3.3. The Role of Food Intake as Symptoms’ Trigger in Lower GI Tract Functional Disorders

#### 3.3.1. The Role of Food Intake as a Symptom Trigger and Therapeutic Target of Carbohydrate Malabsorption Syndromes

Malabsorption refers to the impaired uptake of ingested food components in the gastrointestinal tract. Malabsorption can be caused by various conditions: deficiency of digestive enzymes, inflammation or villous atrophy, altered anatomy following surgery (e.g., small intestinal resection) [42]. In the present review we will focus on the role of deficient digestive enzymes in some DGBIs.

Lactase is the enzyme responsible for hydrolyzation of the β-glycosidic linkage in lactose [43]. While the descendants of northern European populations have high rates of lactase persistence, in most parts of the world enzyme activity does not persist after weaning from breastfeeding—thus approximately 68% of the worlds adult population present lactose malabsorption [44]. While malabsorption of lactose is the norm in most parts of the world, not all individuals with malabsorption also present lactose intolerance, i.e., symptoms such as bloating and abdominal pain following the ingestion of lactose [35]. Conversely, individuals with lactose malabsorption tolerate small amounts of lactose upon blinded exposure [45]. These observations suggest that factors other than the pure malabsorption of lactose contribute to symptom generation, for instance visceral hypersensitivity.

Sucrase-isomaltase (SI) is a brush border enzyme required to hydrolyze α-glycosidically linked polysaccharides such as starch and sucrose thus enabling their absorption [46]. Recent data have shown a higher prevalence of single nucleotide polymorphisms (SNPs) conferring reduced SI enzyme activity in patients suffering from IBS versus healthy individuals [47]. Symptomatic improvement was found when IBS patients followed starch and sucrose restricted diet—However, this needs to be interpreted with caution as one trial was uncontrolled [48] and another study found high response rates to the diet in both patients with and without SI genetic variants [49]. Further research is necessary to prospectively identify these patients and further elucidate the role of dietary interventions in these patients—while a starch reduced diet may be beneficial, a low FODMAP diet has been found to be less effective in a small cohort of patients with decreased SI activity [50] (Figure 1).

In conclusion, while lactose malabsorption in adults is a physiological situation, some individuals experience symptoms—symptom generation is most likely multifactorial and might involve local reactions to milk proteins and visceral hypersensitivity. Reduced enzyme activity of sucrase isomaltase might contribute to symptom generation in some IBS patients but more research is needed to further elucidate the role of reduced enzyme activity and possible dietary interventions in this population.

#### 3.3.2. The Role of Food Intake as a Symptom Trigger and Therapeutic Target of Irritable Bowel Syndrome

According to the Rome IV criteria, IBS is characterized by recurrent abdominal pain, present at least one day a week in the last three months and associated with two or more of the following: related to defecation, associated with change in frequency of stool, and/or change in form of stool [51]. Based on the Bristol stool form, patients with IBS can be subdivided into IBS predominant constipation (IBS-C), IBS predominant diarrhea (IBS-D), IBS mixed stool type (IBS-M) and IBS unclassified (IBS-U). The pathophysiology of IBS is heterogeneous and unclear, but recent observations support a major role for food in IBS symptom onset. An online survey, filled out by 1012 Belgian subjects, identified dietary factors as triggers for their symptoms in 40% of the subjects [52]. This was in line with previous research where a standardized meal test was shown to increase gastrointestinal symptoms in patients with IBS, but not in healthy subjects [53]. Colomier et al. confirmed the association between food intake and gastrointestinal symptoms [54]. In this study, energy intake was correlated with symptom severity and fiber intake with stool consistency. In a diary study of Clevers et al., associations between food and symptoms in patients with IBS were also found [55]. They showed a link between late-night eating and symptoms as abdominal pain, bloating, gas, and nausea.

A dietary intervention with established efficacy in improving gastrointestinal symptoms in patients with IBS, is based on the elimination of fermentable oligo-, di-, monosaccharides and polyols (FODMAPs). This includes fructose surpassing glucose levels, oligosaccharides (both galacto-oligosaccharides and fructans), polyols (mannitol, sorbitol) and the disaccharide lactose. FODMAPs are not effectively absorbed in the small intestine, have an osmotic effect, and are fermented by the gut microbiota. The amount of hydrogen and methane produced by FODMAPs was higher in patients with IBS compared to healthy volunteers [56]. When using magnetic resonance imaging (MRI) to evaluate the gas content in the bowel, fructose, but not inulin, resulted in distention of the small intestine. The combination of glucose and fructose had less effect on the small bowel water content and colonic gas compared to fructose alone, which is in agreement with known fructose co-transport absorption [57] (Figure 1).

Due to these actions, the intake of FODMAPs result in symptoms through heightened water and gas presence in the intestinal tract [58]. However, the mechanism behind FODMAPs is incompletely understood and other factors might play a role. A low FODMAP diet showed lower urinary histamine levels compared to a high FODMAP diet, but further research is needed to evaluate the effect on histamine [59]. In addition, a low FODMAP diet reduced levels of Bifidobacteria in the stool, indicating that food high in FODMAPs might lead to unfavorable changes in microbiota [60]. FODMAPs also includes polyol sweeteners such as mannitol, which is present in chewing gum. The role of food additives, such as sweeteners, is not well established in IBS. However, there is some literature suggesting that food additives have an impact on both the composition and function of the microbiota [61].

Besides the well-known lactose and fructose (FODMAPs) intolerance, an increased prevalence of sucrase-isomaltase gene variants have been reported in IBS, possibly leading to symptoms with intake of sucrose and starch [62]. Nilholm et al. showed symptom improvement in patients with IBS after following a restricted starch and sucrose diet. Based on this study cohort, sucrase-isomaltase genotype was related to symptom improvement in patients with IBS-D [49].

In addition, it remains unclear if the effect of a low FODMAP diet in improving IBS symptoms is due to the reduction in carbohydrates or other dietary factors. Wheat, known as one of the six main food allergens, comprises carbohydrates and both gluten and non-gluten proteins. Vazquez-Roque et al. showed a significant reduction in stool frequency when comparing a gluten-free and gluten-containing diet [63]. Symptom scores of patients with non-celiac gluten sensitivity and IBS improved after a reduced FODMAP intake but worsened thereafter when patients added gluten or whey protein to their diets [64]. However, no significant differences were observed when comparing high, low gluten and placebo treatment arms, indicating that the effect of gluten on symptom levels is unclear.

It is well established that food sensitivities caused by allergic pathophysiology, either due to immunoglobulin E (IgE) or non-IgE-mediated mechanisms, or their combination, can trigger gastrointestinal symptom [65]. Systemic food allergy has been implicated and food could cause a local immune response in IBS patients, although IgE testing is negative in the vast majority of patients. Confocal laser endomicroscopy (CLE) is a novel technique which allows to assess immediate duodenal mucosal responses and identify triggering food proteins even though the patients lack circulating IgE antibodies to these nutrients [66]. In this non-invasive in vivo technique, a CLE probe is passed through the working channel of a standard gastroscope, to visualize in detail the intestinal epithelium after intravenous administration of fluorescein which is excited by a low energy blue laser. This technology enables visualization of duodenal epithelium on a cellular level, as well as observation of acute extravasation of fluorescein into the lumen, in response to local administration of food protein solutions. Acute mucosal alterations were found in IBS m and IBS-D patients after food administration, but more research is needed to define the pathophysiological mechanism behind it (Figure 1).

Additionally, fiber is fermented in the intestine resulting in the production of short-chain fatty acids and gas, and therefore also leading to symptoms such as pain, bloating, and flatulence. However, soluble fibers which are less susceptible to fermentation, can also be used as a treatment for IBS-C patients [67].

#### 3.3.3. The Role of Food Intake as a Symptom Trigger and Therapeutic Target of Diarrhea

Functional diarrhea (FDr) is a disorder of the gut–brain interaction (DGBI), characterized by frequent, loose, and watery stools in the absence of an organic cause. According to the ROME IV criteria, FDr can be differentiated from diarrhea-predominant irritable bowel syndrome (IBS-D) and functional bloating and distention based on the presence and frequency of abdominal pain or bloating [51], which should not be the predominant symptom in FDr [68,69]. The clinical guidelines of the United European Gastroenterology and European Society for Neurogastroenterology and Motility recognize FDr and IBS-D as two potentially overlapping conditions [53].

The pathophysiology of FDr is complex and not fully understood. Several mechanisms may play a role in the progression of FDr. One of the assumed key mechanisms is altered intestinal motility, which may cause increased transit time or inefficient absorption of water from the stool, resulting in increased stool frequency and watery stools. Frequent bowel movements can lead to nutrient malabsorption, potentially causing malnutrition, micro- and macronutrient deficiencies, dehydration, anemia and changes in the gut microbiome [70].

Dietary factors can trigger symptoms of diarrhea due to the effect on gastrointestinal motility and transit time. Most clinical trials investigating the effect of dietary factors on stool frequency and stool consistency tend to focus on individuals with IBS-D or fecal incontinence (FI). Potential triggers for increased bowel movements and looser stools include high-fat foods, fermentable carbohydrates (such as lactose, fructose and sorbitol), caffeine, spicy foods and alcohol [71,72]. Elimination diets have been suggested to reduce symptoms of FD. Given that FDr is a cause of concern for poor nutrient absorption, and more recently, an association has been found between food avoidance or restrictive eating in patients with disorders of gut–brain interaction [73], a gentler approach is advocated. A food-and-stool diary is recommended to identify specific triggers, to avoid unnecessary restrictions, gain symptom improvement and nutrient adequacy. Dietary advice should be provided by a registered dietitian [55] (Figure 1).

For some patients with FDr, increasing soluble dietary fiber can help regulate bowel movements and reduce diarrhea episodes. Soluble fiber, found in foods like oats, bananas, root vegetables, seeds and nuts can be particularly beneficial as it absorbs water and can help bulk up stool [74]. Fiber intake can also be increased by fiber supplementation products, such as psyllium, gum arabic or methylcellulose. Psyllium has shown to reduce symptoms in patients who have FI as a result of diarrhea [75,76]. Gum arabic reduced FI episodes in a pilot study of 39 patients [60]. However, in a single blind randomized controlled trial of 209 patients, supplementation with Arabic gum did not significantly change FI frequency and methylcellulose resulted in an increase in FI [61]. More recently, a randomized pilot study compared the effectiveness of the low FODMAP diet versus psyllium in patients with FI and loose stools. A greater reduction of FI episodes was found with psyllium compared to the low FODMAP diet [77].

In conclusion, food plays an important role in FDr. The main goal is to identify potential dietary triggers, avoid unnecessary restrictions and to provide adequate nutrient intake. Soluble fiber supplementation in the form of psyllium husk may be beneficial to bulk up the stool. Current suggestions of dietary management for FDr are largely aspirational, given the lack of supportive outcomes data. More research is needed to elucidate the causal relationships between food and diarrhea.

#### 3.3.4. The Role of Food Intake as a Symptom Trigger and Therapeutic Target of Constipation

Constipation is a prevalent disorder that has been defined by a European consensus as difficult, unsatisfactory or infrequent defecation typically presenting as the infrequent passing of hard stools and the feeling of incomplete evacuation [78]. Cross sectional studies have shown altered nutritional habits in individuals with versus without constipation. In the general population of Luxembourg, constipation scores correlated negatively with the intake of grains, lipids and positively with total energy intake, sodium intake and consumption of sugary products [79]. Conversely, in 3835 Japanese dietetic students the intake of confectioneries and bread showed a positive correlation with constipation whereas intake of rice and pulses was negatively correlated with constipation [80]. Elderly inhabitants of community centers and nursing homes showed a lower intake of protein, carbohydrates [81]. However, to the best of our knowledge, no longitudinal population-based data are available to decipher causal relationships of food intake and constipation.

While some studies showed that constipated individuals have a lower water intake than non-constipated individuals, four others did not find a relationship between water intake and constipation, and two found a relationship only with water intake from foods and not from fluids [82]. A multimodal education program that also included the instruction to drink 1500–2000 mL of water daily proved to be effective in relieving constipation symptoms. However, most of these individuals had an insufficient water intake at baseline [83].

While some cross-sectional studies found an association of constipation with low fiber intake [65], others did not [63,66] Nevertheless, educating women to increase their fiber intake was included in a multimodal education program shown to be effective [67]. Similarly, various modes of fiber supplementation have been shown to be effective as confirmed in a recent meta-analysis [84]. Dietary supplementation with dried prunes [85,86] improved constipation and was more effective than psyllium—because both interventions were equal in fiber content, the beneficial effect of prunes is likely also mediated by other components such as sorbitol [68]. More recently, prune juice was also found to be effective in treating constipation [87] (Figure 1).

In conclusion, while there is an association of dietary intakes with constipation, the associations found vary depending on the demographic assessed; longitudinal data are necessary to elucidate causal relationships of food intake and constipation. Fiber supplementation with soluble fibers, especially psyllium husk has shown consistent benefits in constipation. Prune juice or dried prunes might provide added beneficial effects mediated by components other than fiber.

#### 3.3.5. The Role of Food and Gut Microbiota Interaction in DGBIs

The gut microbiota play a role in symptom generation in DGBIs. In fact, several treatment approaches modulating gut microbiota composition (e.g., poorly absorbed antibiotics, probiotics and fecal microbiota transplantation) are able to modify the symptoms profile. More in detail, gut microbiota “eubiosis” and its related metabolites are significantly different in IBS vs. healthy subjects [88,89].

Interestingly, food components can interact with the gut microbiota and its metabolites. For example, the intake of FODMAPs and tryptophan can induce the release of neuroactive mediators (e.g., 5-hydroxytryptamine (HT), histamine, proteases, and lipopolysaccharides) that can trigger and increase visceral hypersensitivity through modulation of intestinal nociceptive signaling [90,91]. Further, fecal supernatants of IBS patients were able to cause hypersensitivity in visceral afferents in mice colon. Indeed, this effect was blocked by protease inhibitors and histamine antagonists. On the other hand, the visceral hypersensitivity disappeared when patients followed a low FODMAP diet [92].

#### 3.3.6. The Role of Cognitive Factors in the Interaction between Food and DGBIs

FD and IBS share similar pathophysiological mechanisms and, specifically, psychological comorbidities, such as depression and anxiety disorders, are more frequently detected in these patients compared to the general population [11].

Several studies reported activity in several brain regions to be associated with pain processing in patients with FD vs. healthy controls at resting state and, also after meal ingestion in response to volume of the food [11]. Similarly, there is activation of those brain regions in response to rectal distention in patients with IBS [11]. In detail, the prefrontal cortex (PFC) is the brain area involved in emotion suppression and decision-making. Interestingly, impairment of PFC functioning in IBS and FD patients subjected to hollow visceral ballon dilatation has been reported [11,87].

Fatty food intake can trigger gastrointestinal (GI) symptoms in IBS and FD patients [11,60]. FD patients have a lower rating score for high-fat food images vs. healthy controls [11]. Differently, patients with IBS have a higher consumption of meat [11]. In FD patients, the entry of fat into the duodenum is associated with increasing nausea [11,37].

Using H2 15O-positron emission tomography (PET), PFC was more activated by gastric balloon distention in non-abuse patients with FD than in abuse patients with FD [93]. Further, PFC activation during gastric balloon distention occurs at a lower threshold in patients with FD vs. healthy controls [94]. Thus, PFC hyperactivation is related to aversive stimulation in patients with FD. The dorsolateral PFC (DLPFC) hyperactivation was observed in patients with FD and it has been hypothesized to be associated with cognitive demands and taste-associated activity [94].

## 4. Conclusions

Evidence from literature show a significant association between food ingestion and symptom generation in DGBIs. As outlined above, in FEDs patients, TRPV1 receptors seem to be involved in food hypersensitivity as a 5–6 week treatment with capsaicin, a desensitizing TRPV1 receptor agonist, significantly improved FD symptoms [13].

In PDS patients, early satiation and post-prandial fullness are frequently reported after ingestion of processed food (e.g., with high caloric and fat contents). Moreover, spicy foods, especially those containing capsaicin can elicit FD symptoms. Cognitive factors (e.g., knowledge about food composition by patient) are also involved in symptom generation. Finally, duodenal permeability changes associated with low-grade eosinophilic and mast cell inflammation and activation, similarly to EoE, explain the efficacy of low-protein diets in certain FD patients (e.g., low-FODMAP, gluten free diets).

Brush border digestive enzymes deficiencies are responsible for malabsorptive symptoms, also in a subset of IBS. However, the latter evidence needs more population-based studies. Indeed, IBS subjects report symptom onset upon food ingestion. In detail, energy intake has been correlated with symptom severity and fiber intake with stool consistency. Thus, the elimination of FODMAPs from diet has some evidence but the mechanism behind low-FODMAPs diet effects on IBS is incompletely understood. Further, it remains unclear whether the effect of a low FODMAP diet in improving IBS symptoms is due to the reduction in carbohydrates or other dietary factors (namely, carbohydrates and both gluten and non-gluten proteins). In conclusion, in the framework of pathophysiological understanding of IBS, immunoglobulin E (IgE) or non-IgE-mediated mechanisms have been considered. In particular, confocal laser endomicroscopy usage has shown acute mucosal alterations upon food ingestion in IBS-M and IBS-D patients, but the implications in clinical practice in dietary elimination selection are currently unclear.

In functional diarrhea patients, increased bowel movements and looser stools can be triggered by high-fat foods, fermentable carbohydrates, caffeine, spicy foods, and alcohol. Thus, elimination diets via food diary completion are advocated. Moreover, soluble dietary fiber can reduce diarrhea episodes.

In the case of constipation, no longitudinal population-based data are available in favor of food as trigger factor. However, reduced water and fiber intake have been associated with constipation.

Altogether, this evidence shows different symptom associations with food and its composition. Special attention should be paid also at cognitive factors linking to food ingestion in DGBIs. Longitudinal RCTs are needed to confirm mechanistic, population-based investigations. Therefore, large interventional studies and guidelines for food administration in DGBIs are warranted.

## Figures and Tables

**Figure 1 nutrients-16-00176-f001:**
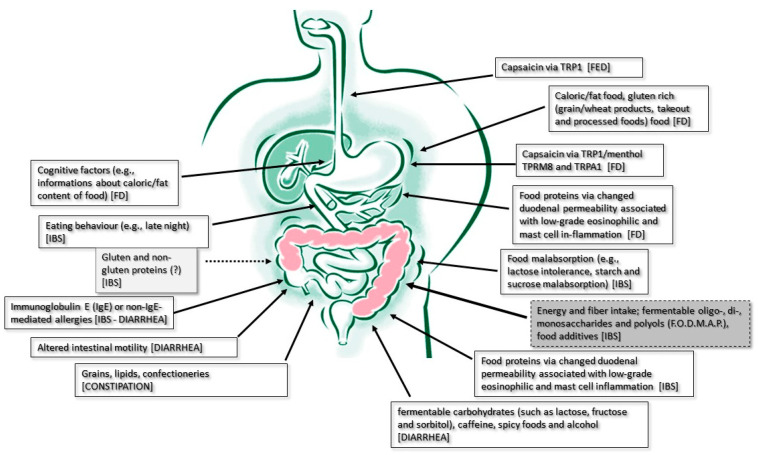
Physiopathological mechanisms at the basis of food role in DGBI symptom generation. In light grey box, the least evidence-based pathophysiological food-symptoms association mechanism; in dark grey box the most evidence-based pathophysiological one.

## Data Availability

All the data reviewed in this manuscript are available online on PubMed and on the database from the main national and international nutrition and gastroenterological meetings (e.g., European Society for Enteral and Parenteral Nutrition, United European Gastroenterology Week, Digestive Disease Week).

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
