# Peer review of "Nutrition and Disorders of Gut–Brain Interaction"

_nutrients, 2024, doi:10.3390/nu16010176_

Round 1
Reviewer 1 Report
Comments and Suggestions for Authors
Congratulations on the article. I enjoyed reading it. I would have one minor objection and that would be for you to try to find and to put in your paper some more information about cognitive factors leading to the which are also involved in disorders of the gut-brain interaction.
Check references.
Comments on the Quality of English LanguageI am satisfied with language.
Author Response
REFEREE 1:
Congratulations on the article. I enjoyed reading it. I would have one minor objection and that would be for you to try to find and to put in your paper some more information about cognitive factors leading to the which are also involved in disorders of the gut-brain interaction.
We thank the review for the appreciation and the suggestion. We have made the adding accordingly.
Check references.
We have checked and corrected the references.
Reviewer 2 Report
Comments and Suggestions for Authors
The effect of diet on health is very often the subject of research. The authors provided an review of 90 selected publication, including original works, reviews and abstracts for determining the role of nutritional factors in the pathogenesis of functional gastrointestinal disorders. The presented material does not contain much new and important information. Knowledge about the main ingredients of food is generally known, but still incomplete. In the summary, authors rightly concluded that further research is necessary. I suggest, to add the purpose of research not only on macronutrients, but also on microelements. In particular, taking into account the involvement of the microbiome and its numerous metabolites in pathogenesis of disorders of gut-brain interaction (DGID).
Editorial comments:
- Complete the research methods section.
- Remove gaps and errors in the references in according with the publisher’s requirements.
Author Response
REFEREE 2:
The effect of diet on health is very often the subject of research. The authors provided an review of 90 selected publication, including original works, reviews and abstracts for determining the role of nutritional factors in the pathogenesis of functional gastrointestinal disorders. The presented material does not contain much new and important information. Knowledge about the main ingredients of food is generally known, but still incomplete. In the summary, authors rightly concluded that further research is necessary. I suggest, to add the purpose of research not only on macronutrients, but also on microelements. In particular, taking into account the involvement of the microbiome and its numerous metabolites in pathogenesis of disorders of gut-brain interaction (DGID).
Editorial comments:
- Complete the research methods section.
- Remove gaps and errors in the references in according with the publisher’s requirements.
We thank the reviewer for the observations and suggestions.
In detail, we have added material on the involvement of the microbiome and its metabolites in the pathogenesis of disorders of gut-brain interaction (DGIB).
We have completed the research methods section.
Finally, we have removed gaps and errors in the references in according with the publisher’s requirements.
Reviewer 3 Report
Comments and Suggestions for Authors
An excellent revision, structurally well designated, providing comprehensive information in a highly usable way.
Since the stated target is DGBIs of the upper and lower GI tracts, I also expect inclusion in the review of functional disorders of the esophagus
Author Response
REFEREE 3:
An excellent revision, structurally well designated, providing comprehensive information in a highly usable way.
Since the stated target is DGBIs of the upper and lower GI tracts, I also expect inclusion in the review of functional disorders of the esophagus.
We thank the reviewer for the suggestion. We have added some data (from the very limited literature on this aspect) on functional disorders of the esophagus. Therefore, we have also updated the figure.
Reviewer 4 Report
Comments and Suggestions for Authors
The effects of diet on one's health are often the focus of studies, thus a review such as this is crucial in providing summary of works that investigated the role of nutrients in the pathogenesis of gastrointestinal disorders. While there exists no special new information in this manuscript, the authors did connect various important published knowledge on the subject, to enhance our understanding of the role of food intolerance in gastrointestinal disorders. I suggest that a few modifications can be made to improve the manuscript.
Minor Revisions
1. The materials and method section needs much elaboration. It should be revised to include specifics; e.g. how many articles were retrieved? What categories could these articles be grouped into?
2. Revisions can be made such that associations can be made between the retrieved manuscripts and their findings, then conclusion could be made based on the number articles that agree with the different symptom that is related to the food composition.
3. If these changes are not to be made, then the entire materials and methods as well as results section should be removed. The manuscript should be written as a traditional review article which should not have Materials and Methods sections as well as Results sections.
4. Subtitles are not good numbered. Please correct.
Author Response
REFEREE 4:
The effects of diet on one's health are often the focus of studies, thus a review such as this is crucial in providing summary of works that investigated the role of nutrients in the pathogenesis of gastrointestinal disorders. While there exists no special new information in this manuscript, the authors did connect various important published knowledge on the subject, to enhance our understanding of the role of food intolerance in gastrointestinal disorders. I suggest that a few modifications can be made to improve the manuscript.
Minor Revisions
- The materials and method section needs much elaboration. It should be revised to include specifics; e.g. how many articles were retrieved? What categories could these articles be grouped into?
We thank the reviewer for the observations and suggestion. We have added to the Methods section more detailed information.
- Revisions can be made such that associations can be made between the retrieved manuscripts and their findings, then conclusion could be made based on the number articles that agree with the different symptom that is related to the food composition.
- If these changes are not to be made, then the entire materials and methods as well as results section should be removed. The manuscript should be written as a traditional review article which should not have Materials and Methods sections as well as Results sections.
- Subtitles are not good numbered. Please correct.
We thank the reviewer for the observation, we have made the changes trying to improve the manuscript. We want to precise that this manuscript has been meant and written as a traditional narrative review of literature. Thus, it has Methods and Results’ sections as it is required. Due to the heterogeneity and complexity of results of food relationship with DGBI’s symptoms, it is not feasible to associate a certain number of articles to some findings in the frame of the Conclusion section. In fact, every reviewed study shows findings that are not only unidirectional and uniform towards a clear and direct food-symptoms association in DGBI patients. The association is variable and multifaceted. For all these reasons, we have summarized the results reviewed and critically evaluated their impact in DGBI pathophysiology and clinics. This process has been finalized also in the Conclusion section.
We have renumbered the paragraph subtitles.